# Sex Differences in Frailty Factors and Their Capacity to Identify Frailty in Older Adults Living in Long-Term Nursing Homes

**DOI:** 10.3390/ijerph20010054

**Published:** 2022-12-21

**Authors:** Nagore Arizaga-Iribarren, Amaia Irazusta, Itxaso Mugica-Errazquin, Janire Virgala-García, Arantxa Amonarraiz, Maider Kortajarena

**Affiliations:** 1Department of Nursing II, Faculty of Medicine and Nursing, University of the Basque Country, 20014 Donostia/San Sebastián, Spain; 2Osakidetza Basque Health Service, Hematology Service, Donostia University Hospital, 20014 Donostia/San Sebastián, Spain; 3Department of Nursing I, Faculty of Medicine and Nursing, University of the Basque Country, 48940 Leioa, Spain; 4Osakidetza Basque Health Service, OSI Tolosaldea, Tolosa Primary Care Center, 20400 Tolosa, Spain; 5San José Long-Term Nursing Home, 20240 Ordizia, Spain

**Keywords:** frailty, capacity to identify, stratified by sex, multidimensional

## Abstract

Frailty is a phenomenon that precedes adverse health events in older people. However, there is currently no consensus for how to best measure frailty. Several studies report that women have a higher prevalence of frailty than men, but there is a gap in studies of the high rates of frailty in older people living in long-term nursing homes (LTNHs) stratified by sex. Therefore, we analyzed health parameters related to frailty and measured their capacity to identify frailty stratified by sex in older people living in LTNHs. According to the Fried Frailty Phenotype (FFP), anxiety increased the risk of frailty in women, while for men functionality protected against the risk of frailty. Regarding the Tilburg Frailty Indicator (TFI), functionality had a protective effect in men, while for women worse dynamic balance indicated a higher risk of frailty. The analyzed parameters had a similar capacity for detecting frailty measured by the TFI in both sexes, while the parameters differed in frailty measured by the FFP. Our study suggests that assessment of frailty in older adults should incorporate a broad definition of frailty that includes not only physical parameters but also psycho-affective aspects as measured by instruments such as the TFI.

## 1. Introduction

Frailty is characterized as increased vulnerability to various adverse health outcomes and is associated with falls, cognitive impairment, disability, hospitalization, institutionalization, and death [1]. There is clear evidence of the clinical relevance of frailty, so it is essential both to identify frail people and to understand the factors and indicators of frailty in order to intervene to prevent adverse consequences [2].

In the last decade, there has been great concern about early detection of frailty in the elderly, considering that aging of the population will increase the number of elderly people with frailty and consequently their functional decline and adverse consequences [3]. The ADVANTAGE Joint Action was developed to address this public concern [4]. It aims to build a common understanding on frailty to manage older people who have or are at risk of developing frailty and to enhance healthy aging.

While frailty is commonly accepted as a multidimensional characteristic of older adults [5], there is less consensus on the appropriate tools to measure frailty. Researchers use multiple scales to assess frailty—all quantify deficits in health but differ in the nature and number of deficits they assess. This difference can be traced to two different conceptual models for frailty scales: the frailty phenotype, and the multidimensional model. The phenotype approaches frailty as a syndrome that comprises specific physical health deficits [6]. On the contrary, the multidimensional model is built on the cumulative deficit approach [7] and assesses frailty as a multidimensional risk condition measured more by the quantity than the nature of health problems [8].

Numerous recently published studies have analyzed or developed instruments to assess and detect early frailty or its adverse effects. For example, the Survey of Health, Ageing and Retirement in Europe (SHARE) published its SHARE Frailty Instrument (FI), which aims to rapidly assess frailty in primary care [9,10,11]. Other studies have provided screening models, algorithms, and recommendations to assess frailty in older people living in different settings [12,13,14]. As frailty assessment is important for clinical care, research, and policy planning, better comprehension of frailty scales in all settings is essential to improve outcomes for older adults [14,15].

Frailty syndrome has been widely researched in older adults in community settings. However, there is less evidence on assessment of frailty in institutionalized older adults [16,17]. This is an important knowledge gap because institutionalized older adults have worse outcomes and more frailty than those in community settings [18,19]. In addition, men and women have different frailty profiles. Several studies report that women have a higher prevalence of frailty than men across all age groups [20]. This is observed both in older people living in the community and in long-term nursing homes (LTNHs) [21,22]. However, women tolerate this frailty better, as evidenced by a lower mortality rate at any level of frailty or age compared to men [17,23]. These findings are in line with the well-described phenomenon that women tend to have poorer health than men but greater longevity [24], termed the health-survival paradox [25].

Whether frailty represents the link between health status and adverse events such as disability, institutionalization, and death in people of the same chronological age [26] may provide a useful paradigm to research the health-survival paradox between men and women to understand why they age differently [20]. This paradox supports the importance of considering sex when addressing frailty and aging interventions. However, most studies do not stratify by sex to analyze differences in frailty transition patterns, factors, and indicators [25]. Therefore, this study aimed to understand how frailty indicators differ in institutionalized men and women. The first objective was to analyze the health parameters related to frailty stratified by sex. The second objective was to measure the capacity of these health parameters to identify the frailty level of men and women living in LTNHs.

## 2. Materials and Methods

### 2.1. Study Design and Participants

This multicenter, cross-sectional study was conducted at 16 LTNHs in Gipuzkoa (Basque Country, northern Spain). Data were collected in January–June 2018. Participants were enrolled based on the following inclusion criteria: age ≥ 70 years; score of ≥50 on the Barthel Index [27]; score of ≥20 on the Mini-Examen Cognoscitivo-35 (MEC-35) test (an adapted and validated version of the Mini-Mental State Examination in Spanish) [28]; and capable of standing up from a chair and walking independently or with any aid for at least 10 m. Exclusion criteria included: if participation was judged inappropriate by a medical expert due to any risk of heart failure or ischemic events; in cases of severe physical, cognitive, or psychiatric disorders; or in cases of any other condition for which study participation was not in the best interest of the individual. After considering these criteria, 199 older adults were included in the study (Figure 1). All participants gave informed consent, and the Research Ethics Committee of the University of the Basque Country approved the study (Human Committee Code M10/2018/171).

### 2.2. Variables Measured

Participants’ baseline measurements—functional status, frailty parameters, physical fitness, psycho-affective status, and quality of life—were assessed at the beginning of the study. To ensure reliability in the measurements, each test or assessment was carried out by the same researcher to avoid inter-rater bias. Frailty was assessed using two measurements: the Fried Frailty Phenotype (FFP) [6], based on physical health deficits; and the Tilburg Frailty Indicator (TFI) [29], based on a multidimensional frailty approach. The FFP assesses the presence of five criteria: unintentional weight loss, weakness, exhaustion, slow gait speed, and low physical activity. One point is given for each criterion, and individuals with a score ≥3 are considered frail (range: 0–5) (Appendix A). The TFI is a validated tool [30] and consists of 15 questions on physical, psychological, and social domains of frailty (Appendix A) The physical domain is assessed with eight items: physical health, unexplained weight loss, difficulties in walking, balance, hand strength, physical tiredness, eyesight, and hearing impairments. The psychological domain is assessed with four items: problems with memory, feeling down, feeling nervous or anxious, and inability to cope with problems. The social domain is assessed with three items: living alone, lack of social relationships, and lack of social support. Eleven items have two response categories (yes or no), while four items have three response categories (yes, sometimes, or no). Scores range 0–15, and individuals scoring ≥5 are considered frail. A detailed description of the scoring is provided in Appendix A.

Functional status was assessed using the Barthel Index [27], which measures 10 activities of daily living: bowels, bladder, grooming, toilet use, feeding, transfer, mobility, dressing, stairs, and bathing. The index ranges 0–100, with a score of 100 indicating complete independence (Appendix A). The assessment was conducted with the person’s main caregiver.

To assess physical fitness, dynamic balance was measured using the validated Timed Up and Go (TUG) test [31]. The participant got up from a chair and walked to a cone located at a distance of 3 m, went around the cone, and walked back to sit down again. The time it took the person to complete this circuit was measured. Physical performance was assessed using the Short Physical Performance Battery (SPPB) [32], the best-known instrument for evaluating physical performance of older people. The SPPB, standardized by the National Institute on Aging, includes three tests: static balance (side-by-side, semi-tandem, and full tandem balance for up to 10 s each), gait speed (timed 4 m walk at a self-selected pace), and five sit-to-stand tests (ability and time needed to stand five times as quickly as possible with arms folded across the chest from a straight-backed chair) (Appendix A).

The Goldberg Anxiety and Depression scales were used to assess psycho-affective status in older adults [33] (Appendix A). Cut-off points were ≥4 for the Anxiety subscale and ≥2 for the Depression subscale. Quality of life was measured with the validated Spanish version of the Quality of Life in Alzheimer’s Disease (QoLAD) scale [34,35], which includes 13 items to assess health status, mood, functional abilities, personal relationships, leisure, economic condition, and life in general. Each item was answered on a Likert scale from 1 (poor) to 4 (excellent) (Appendix A). Scores range 13–52, with a higher score indicating better quality of life.

### 2.3. Statistical Analysis

Statistical analysis was performed with SPSS version 27.0(IBM, Armonk, NY, USA). Continuous variables were expressed as mean ± standard deviation (SD) and categorical variables as frequencies and percentages. The Kolmogorov–Smirnov test was used to analyze whether the variable followed a normal distribution; *p* < 0.05 indicated a nonparametric variable. The Mann–Whitney U test was used to compare the means of two groups, as all variables were nonparametric. Simple logistic regression was performed to evaluate the effect of each parameter on frailty measured with different diagnostic tools. Simple logistic regression is a statistical test used to predict a binary outcome [36]. The variable we wanted to predict was frail or non-frail measured by the FFP or the TFI, and the independent variables were health parameters introduced individually in each simple logistic regression model. From logistic regression, we obtained the odds ratio (OR) to inform the degree of risk estimation or protective effect on frailty. The strength of a diagnostic tool in the case of a binary predictor can be evaluated using measures of sensitivity and specificity. However, in many cases, predictors were measured on a continuous or ordinal range. In these situations, it is desirable to evaluate diagnostic test performance over the range of possible cut-off points for the predictor variable. This is achieved with a receiver operating characteristic (ROC) curve that includes all possible decision thresholds for the outcome of a diagnostic test [37]. Therefore, we analyzed the ROC curves to evaluate the capacity of the parameters (Barthel Index, TUG test, SPPB, Goldberg Anxiety, Goldberg Depression, QoLAD) to identify frailty measured by the FFP and the TFI and to determine an optimal threshold for each parameter to detect frailty incidence. The best cut-off points were determined using the Youden index [38]. The area under the curve (AUC) values of >0.7, >0.8, and >0.9 were considered acceptable, excellent, and outstanding, respectively [37]. All differences were considered significant at *p* < 0.05.

## 3. Results

### 3.1. Participant Characteristics

This study included 199 participants living in 16 LTNHs in Gipuzkoa, Spain (Table 1). The mean age was 85.41 ± 6.51 years. The FFP indicated that 43 men (43%) and 75 women (75.25%) were frail; the TFI indicated that 50 men (50%) and 63 women (63.64%) were frail.

### 3.2. Associations of Health Parameters with Frailty

Simple logistic regression was used for each variable to estimate the degree of risk or protection for frailty (Table 2). For men and women, both the SPPB and QoLAD scale had a protective effect on frailty measured by the FFP. Conversely, the TUG test and Goldberg Depression scale had a risk effect on frailty by the FFP in both men and women. In women, the Goldberg Anxiety scale had a risk effect on frailty by the FFP; in men, the Barthel Index had a protective effect on frailty by the FFP.

For frailty measured by the TFI, both the SPPB and QoLAD scale also showed a protective effect on frailty in men and women (Table 2). In addition, for both men and women, the Goldberg Anxiety and Depression scales had a risk effect on frailty by the TFI. In men, the Barthel Index showed a protective effect on frailty by the TFI; in women, the TUG test showed a risk effect on frailty by the TFI.

### 3.3. Capacity of Health Parameters to Identify Frailty

The AUCs of the ROC curves to identify frail and non-frail individuals according to the FFP and TFI are presented in Table 3 and Figure 2 and Figure 3. Depending on the measure, better health status was signified by the higher ROC curve values (Figure 2a,c and Figure 3a,c) or the lower ROC curve values (Figure 2b,d and Figure 3b,d). Table 4 summarizes and classifies the health parameters with excellent and acceptable capacity to identify frailty according to the AUC for each health parameter and using the classification proposed by Mandrekar [37]. In men, the capability of the physical fitness parameters to identify frail individuals by the FFP was excellent for the TUG test (AUC = 0.840) and SPPB (AUC = 0.802) and was acceptable for the Barthel Index (AUC = 0.764). In women, capability was excellent for the the TUG test (AUC = 0.831) and SPPB (AUC = 0.811). No psycho-affective parameters could identify frail men by the FFP; in women, capability was excellent for the Goldberg Depression scale (AUC = 0.839) and was acceptable for the Goldberg Anxiety scale (AUC = 0.759) and QoLAD (AUC = 0.753).

According to the TFI, no physical fitness parameter could identify frail men or women. The capability of psycho-affective parameters to identify frail men by the TFI was acceptable for the Goldberg Depression scale (AUC = 0.789), Goldberg Anxiety scale (AUC = 0.762), and QoLAD (AUC = 0.705); in women, it was excellent for the Goldberg Depression scale (AUC = 0.858) and was acceptable for the Goldberg Anxiety scale (AUC = 0.782) and QoLAD (AUC = 0.732).

## 4. Discussion

Our data reflect that related health parameters may differ between men and women in their capacity to identify frailty depending on the frailty measurement instrument used (FFP or TFI). Nonetheless, the analyzed health parameters in men and women presented a similar capacity for detecting frailty measured by the TFI, while these parameters differed in the capacity to detect frailty measured by the FFP, which is the most widely used instrument in published studies to date [39]. The reported prevalence of frailty in institutionalized older people in the LTNHs varies depending on the measurement tool used [40], so it is difficult to compare our data with the published data. However, we found a lower percentage of frail men and women when using the FFP compared to the TFI, in agreement with previous publications showing that the use of a definition of physical frailty yields a lower frailty prevalence than a broader frailty definition [41].

Despite recognition of differences between sexes, few studies have stratified frailty data by sex. Some authors have published data stratified by sex of community-dwelling older adults [42,43] while others have included both community-dwelling and institutionalized individuals [44,45,46,47,48,49]. To our knowledge, only two published studies have stratified frailty prevalence data by sex exclusively from a population of older people living in LTNHs [50,51]. Thus, our male and female frailty data add to the scientific literature and confirm these previous studies by showing higher frailty rates in women compared to men.

The FFP is one of the most commonly used measurement instruments in frailty studies of older people [41,52,53,54]. While the FFP is based on the frailty phenotype model, which accounts for the physical components of frailty, the TFI uses the multidimensional model to incorporate a broader definition of frailty by including physical, psychological, and social components [41,55]. We assessed both instruments, representing both models, in our population to more comprehensively analyze the health parameters related to frailty. In agreement with our data, other studies have shown that frailty is related to various factors that affect functionality, physical condition, and even quality of life [56]. We found a positive relationship of the SPPB and QoLAD scale with frailty as well as a negative relationship of the Goldberg Depression scale with frailty according to the FFP and the TFI in both men and women in LTNHs, in agreement with previous studies [50,57,58]. However, we also observed differences between men and women in the relationship of the health parameters with frailty. For instance, the Barthel Index was positively associated with frailty according to the FFP and TFI only in men. Ambagtsheer et al. (2017) also found that some domains of functionality are related to frailty in people living in LTNHs, although they did not differentiate analysis by sex [50]. We observed two results that were specific to women. First, the Goldberg Anxiety score was a risk factor for frailty measured by the FFP, consistent with previous findings from community-dwelling older adults [59]. Second, frailty measured by the TFI was negatively related to the TUG test score in women, as shown in a previous study of institutionalized older adults [57].

When analyzing the capacity of physical health parameters to identify frailty, we also detected similarities and differences between men and women. Regarding similarities between groups, we found that the TUG test and SPPB presented acceptable and excellent capacity for detecting frailty according to the FFP. These results agree with a previously published study analyzing the capacity of different frailty parameters in older people living in LTNHs to identify frailty and showed an excellent capacity of the TUG test, gait speed test, and Berg Balance scale (the latter two of which are included in the SPPB) for frailty according to the FFP [57]. However, studies also have shown the lower capacity of the SPPB to detect frailty. Pritchard et al. [60] showed in a geriatric outpatient population fair to moderate agreement between the FFP and the SPPB to identify frail and pre-frail participants, respectively. Another similar result showed that the SPPB has neither reliability nor validity to measure frailty [54]. Therefore, exclusive use of the SPPB to screen for frailty might be questionable. Regarding differences we observed between men and women, the Barthel Index presented an acceptable capacity to identify frailty only in men, who scored higher than women in the functional test. This finding may suggest that the Barthel Index only has an acceptable capacity to identify frailty in the context of high functionality scores.

For psycho-affective parameters, an acceptable and excellent capacity to identify frailty according to the FFP was achieved only in women. It is surprising that psycho-affective parameters such as the QoLAD and Goldberg Anxiety and Depression scales had high capacity to identify frailty measured with the FFP, a tool focused on only physical parameters. This could be due to the roles that women adopt during their lives and their longer life expectancy, which causes them to be alone in the last years of their lives. This could lead women to have higher levels of anxiety, depression, and loneliness than men, as suggested by Whitesides and Lynn [61]. Therefore, based on these results, we could recommend the use of screening tools that include or complement different frailty, physical, and psycho-affective domains, especially in women. In this sense, the results published by Gilardi et al. (2018) are in line with our results in that they identified the TFI as the best screening tool because it is multidimensional, quick and easy to apply, and has an accurate risk prediction value [55]. Although there are other multidimensional tools that are validated and have a high capacity to identify frailty, the TFI is considered user-friendly to implement [62]. Other recent work also has suggested combining different tools to obtain a higher capacity to identify frailty in older people [49,57]. It might be worthwhile to assess whether combinations of measurements show any significant improvement in the capacity to identify frailty.

For frailty according to the TFI, no physical parameter had an acceptable capacity to identify frailty in men or women. This coincides with results published by Arrieta et al. (2022) showing that the TUG test has a poor capability to identify frailty according to the TFI [57]. However, all psycho-affective parameters presented an acceptable or excellent capacity to identify frailty in both sexes for the TFI. These results are in line with published studies showing that loneliness, engagement in social activities, and apathy could assess or predict frailty in community-dwelling [63,64] and institutionalized [51,57] older people. It is noteworthy that no parameter achieved an outstanding value, but the parameter in our study with best capacity to identify frailty was the Goldberg Depression scale according to the TFI in women. Previous studies have confirmed a relationship between frailty and depression [65], so our results support incorporating psycho-affective parameters when assessing frailty, especially in older women.

## 5. Conclusions

In conclusion, physical, psycho-affective, and quality of life parameters are related to frailty according to the FFP and TFI in both sexes. In addition, our study suggests that assessment of frailty in older adults should incorporate a broad definition of frailty that includes not only physical parameters but also psycho-affective aspects as measured by instruments such as the TFI, which is a quick and simple tool to apply in both clinical and public health settings [41,62]. According to Rodriguez-Mañas et al. [1], the variety of frailty measurement approaches reflects a lack of consensus on what should be considered as the necessary domains or parameters to include when assessing frailty. In this sense, our results provide important information and reflect the need to analyze older men and women residing in LTNHs in an individualized manner and to combine assessment of several domains to detect frailty accurately and reliably. Thus, more specific studies are needed to inform the most effective interventions to prevent or reverse health problems in institutionalized older people.

### Strength and Limitations

The main strength of this study is that the sample represents a large percentage of older people living in LTNHs, both in terms of sample size as well as age and sex characteristics. The tools for assessment of the health parameters (physical capacity, quality of life, and psycho-affective parameters) are widely validated and recognized for use in older people. Two frailty measurement instruments that assess different domains were used to provide a multidimensional analysis of frailty. This is the first study in which the capacity of different health parameters to identify frailty was analyzed separately in men and women. This opens up new directions for future studies in this field.

A limitation of this study is that the results cannot be applied to all residents of LTNHs since there are institutionalized people with worse physical, psycho-affective, and quality of life conditions than included in our sample. Another limitation is that our sample may represent region-specific findings since all LTNHs were in Spain. Additional longitudinal studies are needed to reinforce the hypothesis proposed in this cross-sectional study. In addition, as there is no consensus on the tools to assess frailty or on the domains they should include [48,52], knowledge in this area should be expanded with more studies analyzing frailty data by sex. Therefore, our results should be interpreted with caution.

## Figures and Tables

**Figure 1 ijerph-20-00054-f001:**
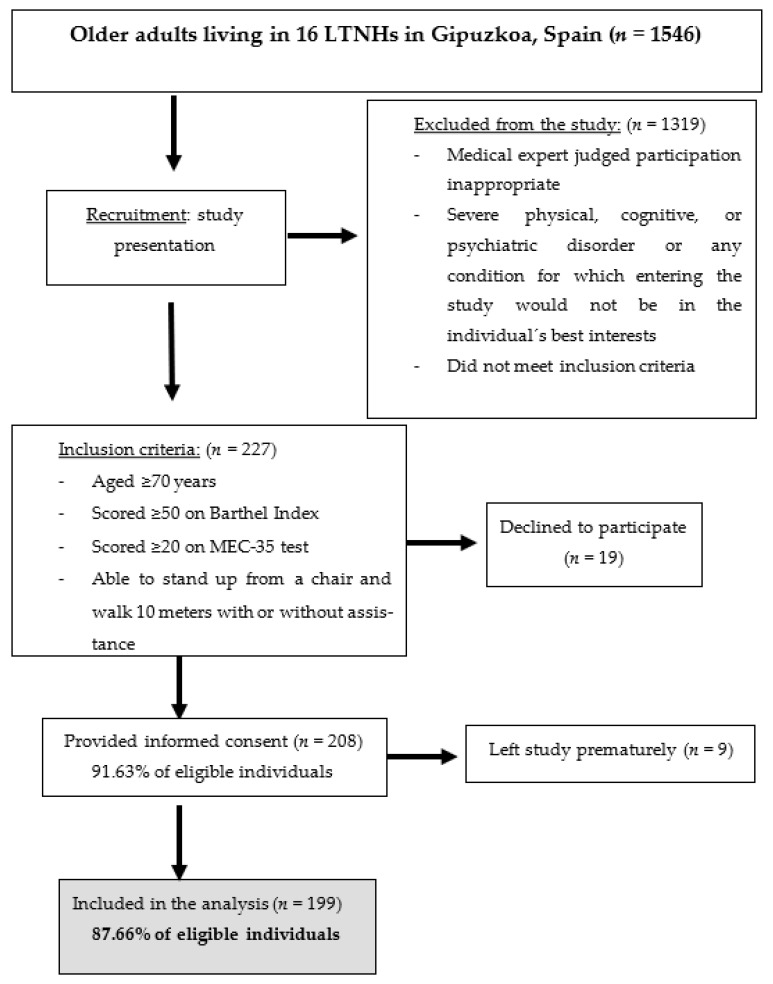
Flow diagram of study selection.

**Figure 2 ijerph-20-00054-f002:**
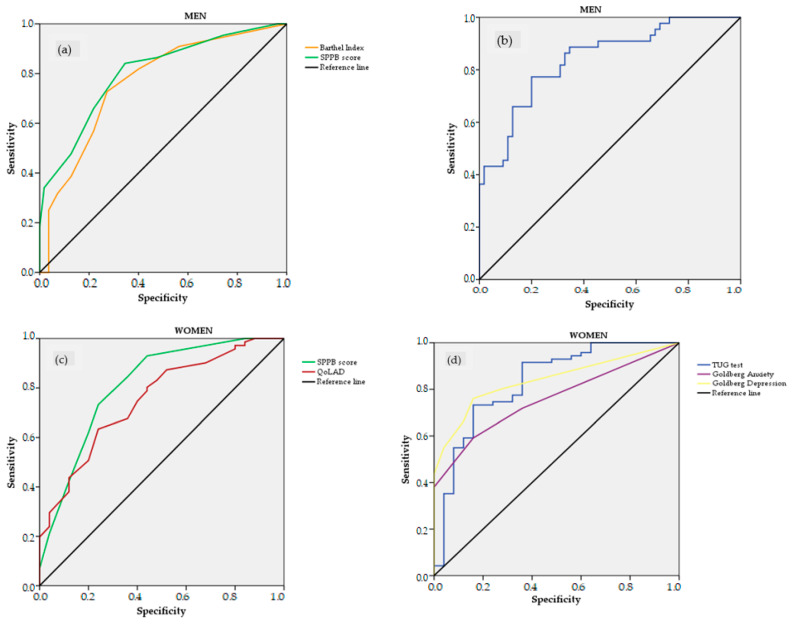
Receiver operating characteristic (ROC) curves to identify frailty by Fried Frailty Phenotype (FFP) in men (**a**,**b**) and women (**c**,**d**) using different measures: Barthel Index, Timed Up and Go (TUG) test, Short Physical Performance Battery (SPPB) score, Quality of Life in Alzheimer’s Disease (QoLAD), and Goldberg Anxiety and Depression scales. ROC curves of TUG test (for men and women) and Goldberg Anxiety and Depression scales (for women) have been plotted inverted for better comparison.

**Figure 3 ijerph-20-00054-f003:**
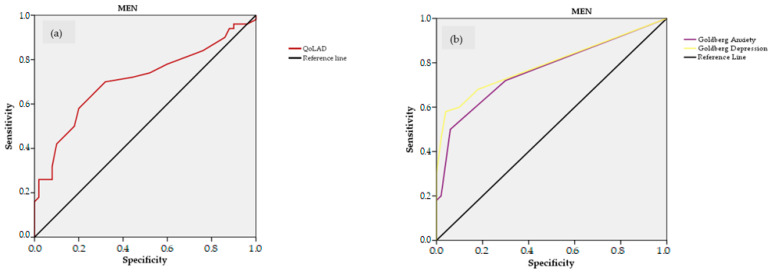
Receiver operating characteristic (ROC) curves to identify frailty by Tilburg Frailty Index (TFI) in men (**a**,**b**) and women (**c**,**d**) using different measures: Quality of Life in Alzheimer’s Disease (QoLAD), and Goldberg Anxiety and Depression scales. ROC curves of Goldberg Anxiety and Depression scales have been plotted inverted for better comparison.

**Table 1 ijerph-20-00054-t001:** Participant characteristics.

	Total (Mean ± SD) *n* = 199	Men (Mean + SD) *n* = 100	Women (Mean + SD) *n* = 99
Age (years)	85.41 ± 6.51	84.24 ± 6.83	86.60 ± 5.97
Barthel Index (range: 0–100)	80.38 ± 15.13	85.95 ± 13.57 ***	74.75 ± 14.59
TUG test (s)	26.10 ± 14.41	22.44 ± 13.71 ***	29.80 ± 14.22
SPPB (range: 0–12)	6.04 ± 2.71	7.04 ± 2.51 ***	5.02 ± 2.52
Goldberg Anxiety (range: 0–9)	1.73 ± 2.40	1.16 ± 1.77 **	2.32 ± 2.80
Goldberg Depression (range: 0–9)	2.30 ± 2.64	1.59 ± 2.20 ***	3.02 ± 2.85
QoLAD (range: 13–52)	32.19 ± 6.47	33.69 ± 5.82 ***	30.63 ± 6.75
FFP score (range: 0–5)	2.77 ± 1.37	2.25 ± 1.26 ***	3.29 ± 1.28
Frail by FFP, n (%)	117 (58.79)	43 (43)	75 (75.25)
TFI score (range: 0–15)	5.28 ± 3.11	4.64 ± 2.70 **	5.90 ± 3.38
Frail by TFI, n (%)	113 (56.78)	50 (50)	63 (63.64)

FFP = Fried Frailty Phenotype; QoLAD = Quality of Life in Alzheimer’s Disease; SD = standard deviation; SPPB = Short Physical Performance Battery; TFI = Tilburg Frailty Indicator; TUG = Timed Up and Go. ** Mann–Whitney U test *p* < 0.01. *** Mann–Whitney U test *p* < 0.001.

**Table 2 ijerph-20-00054-t002:** Simple logistic regression for frailty measures, including parameters analyzed as independent variables stratified by sex.

	Fried Frailty Phenotype (FFP)		Tilburg Frailty Indicator (TFI)	
	Men OR (95% CI)	Women OR (95% CI)	Men OR (95% CI)	Women OR (95% CI)
Barthel Index	0.930 (0.896–0.965) *	0.987 (0.956–1.019)	0.947 (0.916–0.980) *	0.975 (0.947–1.005)
TUG test	1.205 (1.112–1.305) *	1.143 (1.066–1.225) *	1.025 (0.992–1.060)	1.047 (1.007–1.088) *
SPPB	0.570 (0.450–0.723) *	0.572 (0.440–0.735) *	0.836 (0.708–0.986) *	0.779 (0.652–0.931) *
Goldberg Anxiety	1.214 (0.953–1.546)	2.000 (1.254–3.189) *	3.040 (1.742–5.306) *	2.063 (1.373–3.100) *
Goldberg Depression	1.265 (1.042–1.536) *	2.043 (1.429–2.921) *	2.248 (1.580–3.200) *	1.995 (1.498–2.659) *
QoLAD	0.914 (0.849–0.984) *	0.866 (0.803–0.935) *	0.869 (0.802–0.942) *	0.885 (0.825–0.950) *

CI = confidence interval; OR = odds ratio; QoLAD = Quality of Life in Alzheimer’s Disease; SPPB = Short Physical Performance Battery; TUG = Timed Up and Go. * *p* < 0.05.

**Table 3 ijerph-20-00054-t003:** Sex differences in frailty according to receiver operating characteristic curves.

	**Fried Frailty Phenotype (FFP)**													
	**Men**							**Women**						
	**AUC**	**Cut-Off Point**	**Youden Index**	**Sensitivity**	**Specificity**	***p*-Value**	**95% CI**	**AUC**	**Cut-Off Point**	**Youden Index**	**Sensitivity**	**Specificity**	***p*-Value**	**95% CI**
Barthel Index	0.764	87.50	0.442	0.714	0.727	<0.001	0.670–0.859	-	-	-	-	-	-	-
TUG test	0.840	20.68	0.573	0.800	0.773	<0.001	0.763–0.918	0.831	24.83	0.572	0.840	0.732	<0.001	0.731–0.931
SPPB	0.802	7.50	0.495	0.841	0.655	<0.001	0.715–0.890	0.811	5.50	0.499	0.739	0.760	<0.001	0.705–0.917
Goldberg Anxiety	-	-	-	-	-	-	-	0.759	1.5	0.432	0.840	0.592	<0.001	0.663–0.854
Goldberg Depression	-	-	-		-	-		0.839	1.5	0.601	0.840	0.761	<0.001	0.761–0.918
QoLAD	-		-	-	-	-	-	0.753	29.50	0.422	0.662	0.760	<0.001	0.646–0.860
		**Tilburg Frailty Indicator (TFI)**												
	AUC	Cut-off point	Youden index	Sensitivity	Specificity	*p*-value	95% CI	AUC	Cut-off point	Youden index	Sensitivity	Specificity	*p*-value	95% CI
Barthel Index	-	-	-	-	-	-	-	-	-	-	-	-	-	-
TUG test	-	-	-	-	-	-	-	-	-	-	-	-	-	-
SPPB	-	-	-	-	-		-	-	-	-	-	-	-	-
Goldberg Anxiety	0.762	1.50	0.430	0.940	0.490	0.016	0.666–0.858	0.782	1.50	0.423	0.794	0.629	0.005	0.692–0.872
Goldberg Depression	0.789	0.50	0.493	0.820	0.673	0.005	0.697–0.882	0.858	1.50	0.617	0.794	0.823	<0.001	0.782–0.934
QoLAD	0.705	34.50	0.380	0.700	0.680	<0.001	0.601–0.810	0.732	29.50	0.386	0.672	0.714	<0.001	0.632–0.831

AUC = area under the curve; CI = confidence interval; QoLAD = Quality of Life in Alzheimer’s Disease; SPPB = Short Physical Performance Battery; TUG = Timed Up and Go.

**Table 4 ijerph-20-00054-t004:** Criterion validity for frailty measures.

	Fried Frailty Phenotype (FFP)		Tilburg Frailty Indicator (TFI)	
	Men	Women	Men	Women
**Excellent**	TUG test SPPB score	TUG test SPPB score Goldberg Depression	-	Goldberg Depression
**Acceptable**	Barthel Index	QoLAD Goldberg Anxiety	QoLAD Goldberg Anxiety Goldberg Depression	QoLAD Goldberg Anxiety

QoLAD = Quality of Life in Alzheimer’s Disease; SPPB = Short Physical Performance Battery; TUG = Timed Up and Go.

## Data Availability

The data presented in this study are available on request from the corresponding author.

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
