# Peer review of "Sex Differences in Frailty Factors and Their Capacity to Identify Frailty in Older Adults Living in Long-Term Nursing Homes"

_ijerph, 2022, doi:10.3390/ijerph20010054_

Round 1

Reviewer 1 Report

This study reported the frailty for the people living in long-term nursing homes uisng different indices. They are particularly interested in the people having difference sex. Here are my comments.

1. How to calculate the odds ratio in this work? What is the odds ratio informing? Why are some parameters, such as the Barthel index of women OR and the Goldberg anxiety of men OR, not shown in Table 2?

2. A definition of the simple logistic regression should be given in the paper.

3. What is the predictive value in this study? How was the model established?  I am not sure if the title if this manuscript should be slightly changed to accurately describes the contents of your work. Was a new way to perform the prediction presented?

4. How to establish the ROC curves and can the results be interpreted?

5. How to choose variables that affect FFP and TFI (Table 4)? What is the considered statistical values? How to interpret that considered statistical values?

Reviewer 2 Report

Comments and Suggestions for Authors: 

The paper deals with the serious problem of frailty amongst older people which significantly affects their lives. Specifically, the authors analyze health parameters related to frailty stratified by sex as there are evidently some differences between men and women.  Also, the authors analyze different conceptual models for frailty scales and predictive validity of different health parameters.

The results of the study are presented quite well, the conclusions are clearly justified and correspond to the obtained results. 

There is a minor flaw:

-  For the better perception of the obtained results it is advisable to give a recommended algorithm for assessing frailty in older people. 

Reviewer 3 Report

I feel happy to review this manuscript titled as "Sex Differences in Frailty Factors and Their Predictive Value for Frailty in Older Adults Living in Long-Term Nursing Homes". I have some suggestion for the authors. I hope it will help authors to revise their manuscript.

1. The introduction section having lack of novelty. I suggest authors to add more research contribution in this section.

2. There is no literature review section in this manuscript. I suggest authors to add this section.

3. I suggest authors to add a sample items of survey in the variable measured section.

4. Does authors have tested the reliability of the measurement scale?

5. I suggest authors to add appendix in the revised manuscript. 

Good Luck

Round 2

Reviewer 1 Report

The authors have made efforts to improve the manuscript. I would like to this manuscript is accepted for publication as its current form.

Reviewer 3 Report

I am satisfied with the author response.